# The Protective Role of Job Control/Autonomy on Mental Strain of Managers: A Cross-Sectional Study among Wittyfit’s Users

**DOI:** 10.3390/ijerph19042153

**Published:** 2022-02-14

**Authors:** Rémi Colin-Chevalier, Bruno Pereira, Amanda Clare Benson, Samuel Dewavrin, Thomas Cornet, Frédéric Dutheil

**Affiliations:** 1CNRS, LaPSCo, Physiological and Psychosocial Stress, University Hospital of Clermont-Ferrand, Preventive and Occupational Medicine, Université Clermont Auvergne, Wittyfit, 63000 Clermont-Ferrand, France; fred_dutheil@yahoo.fr; 2Biostatistics Unit, Clinical Research and Innovation Direction, University Hospital of Clermont-Ferrand, 63000 Clermont-Ferrand, France; bpereira@chu-clermontferrand.fr; 3Department of Health and Biostatistics, Swinburne University of Technology, Hawthorn, VIC 3122, Australia; abenson@swin.edu.au; 4Wittyfit, 75000 Paris, France; samuel.dewavrin@wittyfit.com (S.D.); thomas.cornet@wittyfit.com (T.C.)

**Keywords:** job demand, job control, job support, job strain, isostrain, manager, Karasek’s model, work

## Abstract

Background: Karasek’s Job Demand-Control-Support model is the gold standard to assess the perception of work; however, this model has been poorly studied among managers. We aimed to explore the perception of work (job demand, control, and support) in managers, and to quantify their risk of job strain (high job demand and low job control) and isostrain (job strain with low job support). Methods: We conducted a cross-sectional study on workers from various French companies using the Wittyfit software. Job demand, control, and support were evaluated by self-reported questionnaires, as well as sociodemographic data. Results: We included 9257 workers: 8488 employees (median age of 45 years, median seniority of 10 years, 39.4% women) and 769 managers (463 were more than 45 years old, 343 with more than 10 years of service, 33.3% women). Managers had higher mean ± SD levels than employees in job control (79.2 ± 14.9 vs. 75.4 ± 16.9) and job support (25.2 ± 5.1 vs. 24.0 ± 6.1) (*p* < 0.001). Compared to employees, managers had a 37% decreased risk of job strain (OR = 0.63, 95% CI 0.52 to 0.77) and a 47% decreased risk of isostrain (OR = 0.53, 95% CI 0.40 to 0.69) (*p* < 0.001). Workers over age 45 (OR = 1.26, 95% CI 1.14 to 1.40, *p* < 0.001) and women (OR = 1.12, 95% CI 1.01 to 1. 25, *p* = 0.03) were at greater risk of job strain. Furthermore, workers over age 45 (OR = 1.51, 95% CI 1.32 to 1.73, *p* < 0.001), workers with over 10 years of service (OR = 1.35, 95% CI 1.16 to 1.56, *p* < 0.001), and women (OR = 1.15, 95% CI 1.00 to 1.31, *p* = 0.04) were at greater risk of isostrain. Conclusions: Managers seem to have higher autonomy and greater social support and therefore are less at risk of job strain or isostrain than employees. Other factors such as age, seniority, and sex may influence this relationship. Trial Registration: Clinicaltrials.gov: NCT02596737.

## 1. Introduction

Karasek’s Job Demand-Control-Support (JDCS) model [1,2,3,4,5] is a theory that helps to explain the relationship between job characteristics and the psychological well-being of workers. The model explains how the interaction of job demand (or psychological demand, e.g., work overload or role conflict), job control (or decision latitude, e.g., control over work pace or even workplace autonomy), and job support (or social support, e.g., emotional, physical, physiological, or practical assistance) can cause mental strain in individuals [1,2,6,7]. Job strain results from the interaction between high job demand and low job control, and isostrain is characterized by job strain with low social support. Job strain and isostrain have emerged as major public health problems with hazardous effects on workers [8,9,10,11,12,13,14,15,16,17,18,19,20].

Several factors such as age, seniority, or even sex are known to have an impact on job strain among workers [21,22,23,24]. To the best of our knowledge, there are few studies on the links between one’s position within a company (manager or employee) and job strain or isostrain risk [25,26], or studies which were carried out in one industry only, such as healthcare [15,27]. Despite managers’ increased risk of stress [28] due to the multitude of tasks and responsibilities they face [29], their autonomy may overcome their job demand. As managers occupy a crucial role in a company, it therefore seems necessary to evaluate the impact of managerial status on the risk of mental strain, but also to identify other potential factors which may affect this relationship.

Thus, the main objective of our study was to evaluate the influence of job position (manager or employee) on the perception of work (job demand, job control, and job support) and mental strain (job strain and isostrain), particularly the putative protective role of job control among managers. The secondary objectives were to investigate other sociodemographic factors that may influence participant responses, using Karasek’s model, and to quantify the impact of all these factors.

## 2. Materials and Methods

### 2.1. Recruitment

Wittyfit software is a web platform designed to promote and improve well-being in the workplace, implemented in various French industries that are clients of the Wittyfit company (Paris, France) [30]. Any worker wishing to participate in the Wittyfit study could register for free and access the eponymous application, which can be downloaded to a desktop computer or smartphone. Volunteer workers were asked to answer various questionnaires. They could anonymously provide personal information about their physical, mental, and work-related feelings in order to participate in the improvement of their own well-being. Managers and employees received personal and individualized feedback that could be used to implement a preventive strategy. The Wittyfit software was designed in partnership with the University Hospital of Clermont-Ferrand. All companies and workers using Wittyfit on a voluntary basis were included in the study. We limited the inclusion of data up to February 2020, prior to the COVID-19 pandemic.

### 2.2. The Job Demand-Control-Support Model

The three measurements of Karasek’s model, namely, perceived job demand, job control, and job support, were assessed through the Wittyfit platform using visual analogical scales (VASs). VASs are widely used to assess perceived stress in workers [31,32,33]. Wittyfit’s VASs are graded from 0 to 100, with 0 indicating a low level and 100 a high level. Therefore, job demand and job control were, respectively, assessed using the Wittyfit’s VASs “workload” and “autonomy” from the “stress” questionnaire. Job support was assessed by the “ambiance” VAS from the “job satisfaction” questionnaire, designed to assess social support at work [30]. Then, the values were rescaled to match the true measurement scales. From a score of 0 to 100, the levels of job demand, job control, and job support were rescaled to match the true measurement scales of the Karasek questionnaire. Thus, the VAS from 0 to 100 for job demand was converted into a score from 9 to 36, job control into a score from 24 to 96, and job support into a score from 8 to 32. From this conversion, we were able to apply thresholds to our variables to transform them into qualitative variables to determine whether a worker was in a job strain or isostrain situation. Workers with more than 21 out of 36 in job demand and less than 70 out of 96 in job control were considered to be in a job strain situation. Job strain workers with less than 23 out of 32 in job support were considered to be in an isostrain situation. These values were chosen in accordance with the literature [6]. Throughout the duration of the study, when logging back into the platform, the workers could, if they wished, re-express how they felt in the different categories. The last scores entered by the user were used for the study.

### 2.3. Sociodemographic Charactertistics of Workers

Sociodemographic characteristics considered were age, seniority, sex, position (manager or employee), and company they work for. A manager was defined as a worker with primary responsibility, having a leadership position over a group of workers, as opposed to an employee. This information was provided by Wittyfit’s client companies and stored in the Wittyfit database. Thus, for each worker, age, seniority (in age groups), sex (male or female), and position (manager or employee) were known. Age was transformed from a multiclass variable into a binary variable indicating whether the worker was over or under 45 years old. In the same way, seniority was transformed into a binary variable indicating whether the worker had over or under 10 years of service.

### 2.4. Statistical Analyses

Quantitative data (job demand, control, and support), expressed by the mean ± standard deviation (SD), were normalized using a “refit” method [34] and were compared between independent groups (by age, seniority, sex, and job position) with ANOVA. Qualitative data (job strain and isostrain) were described by a number of participants, and associated frequencies were compared between groups with the χ^2^ test. Then, multivariate analysis was conducted using linear or logistic mixed regressions depending on the outcome’s nature (quantitative or qualitative, respectively) to determine variables associated with different Karasek model outcomes. The “company” effect was included as a random effect. Model assumptions (residual independence and normality, and variance homogeneity) were verified a posteriori. Multicollinearity of covariates was checked prior to analysis. Unless specified, all estimates were interpreted in terms of effect sizes (ESs) or standardized odds ratios (ORs) and 95% confidence intervals (95% CIs). ESs were interpreted according to Funder’s rules [35]. Statistical analyses were performed with R (version 4.0.4) [36] in the RStudio (version 1.3.1056) platform. A *p*-value < 0.05 was considered statistically significant for all analyses.

## 3. Results

### 3.1. Participants

Data of 15,562 workers were collected between January 2018 and February 2020. We obtained complete data from 5656 men and 3601 women, i.e., 9257 workers, including 769 managers (8.3% of workers) (Figure 1).

Table 1 shows the sociodemographic characteristics of workers by score obtained using Karasek’s model.

Of the 9257 workers included in the study, 5537 (59.8%) had an “active” job (high demand, high control) according to the Karasek model, 558 (6.0%) a “passive” job (low demand, low control), 927 (10.0%) a “low strain” job (low demand, high control), and 2235 (24.1%) a “high strain” job and were in a job strain situation (high demand, low control). Further, among job strain workers, 1192 individuals (12.9% of the total population) were in an isostrain situation.

### 3.2. Impact of Sociodemographic Characteristics of Workers on Responses to the JDCS Model

#### 3.2.1. Job Demand, Control, and Support among Workers

On average, managers had significantly higher job control (79.2 ± 14.9 vs. 75.4 ± 16.9, *p* < 0.001) and job support (25.2 ± 5.1 vs. 24.0 ± 6.1, *p* < 0.001) than employees, but there was a non-significant difference in job demand (26.8 ± 6.4 vs. 26.7 ± 6.3, *p* = 0.3) (Figure 2). According to our results, both managers and employees using the Wittyfit software felt, on average, “active” at work (Figure 3).

#### 3.2.2. Job Strain and Isostrain Prevalence by Job Position

Managers had a lower risk of job strain (OR = 0.63, 95% CI 0.52 to 0.77, *p* < 0.001) and isostrain (OR = 0.53, 95% CI 0.40 to 0.69, *p* < 0.001) than employees. Thus, there was a lower prevalence of job strain (17.9% of managers vs. 24.7% of employees, *p* < 0.001) and isostrain (8.1% of managers vs. 13.3% of employees, *p* < 0.001) among managers (Figure 4).

#### 3.2.3. Sociodemographic Factors of Karasek’s Model

Age emerged as a factor that influenced some dimensions of Karasek’s model among all workers (both managers and employees). Indeed, workers over 45 had, on average, higher job demand (ES = 0.06, 95% CI 0.01 to 0.10, *p* = 0.009), lower job control (ES = −0.06, 95% CI −0.11 to −0.02, *p* = 0.004), and lower job support (ES = −0.19, 95% CI −0.24 to −0.15, *p* < 0.001). Moreover, similar to age, workers with over 10 years of seniority reported having higher job demand (ES = −0.15, 95% CI −0.20 to −0.10, *p* = 0.02) and less job support (ES = −0.21, 95% CI −0.26 to −0.16, *p* < 0.001) (Figure 5).

Furthermore, women reported having higher job demand (ES = 0.05, 95% CI 0.01 to 0.09, *p* = 0.02) than men but lower job support (ES = −0.10, 95% CI −0.14 to −0.06, *p* < 0.001). Finally, our results show that the prevalence of job strain was higher among over-45 workers (OR = 1.26, 95% CI 1.14 to 1.40, *p* < 0.001) and women (OR = 1.12, 95% CI 1.01 to 1.25, *p* = 0.03). Likewise, the prevalence of isostrain was higher among over-45 workers (OR = 1.51, 95% CI 1.32 to 1.73, *p* < 0.001), workers with 10 or more years of service (OR = 1.35, 95% CI 1.16 to 1.56, *p* < 0.001), and women (OR = 1.15, 95% CI 1.00 to 1.31, *p* = 0.04) (Figure 6).

## 4. Discussion

The main results show that, due to higher job control and job support, managers were at less risk of job strain and isostrain than employees. However, other characteristics such as age, seniority, or sex were found to have an impact on the JDCS dimensions [5], sometimes protective, sometimes hazardous.

### 4.1. The Managers and Karasek’s Model

Among workers who answered the JDCS questionnaire, we found that managers were less exposed to the risk of job strain and isostrain than employees. Indeed, only 17.9% of managers were in a job strain situation, compared to 24.7% of employees. Likewise, only 8.1% of managers were in an isostrain situation, compared to 13.3% of employees. In particular, the results highlight that the high levels of job control and job support perceived by the managers protect them from these risks. This suggests that the role of the manager, and more specifically the specificities of the manager’s work, as opposed to that of the employee, acts as a protective factor against job strain and isostrain. These results confirm previous findings showing the impact of socioeconomic status on workers’ type of job, on the one hand, and on the prevalence of job strain and isostrain, on the other, among workers, particularly the lower prevalence for workers with a higher socioeconomic position such as managers [25,26,37].

We did not show a difference in job demand between managers and employees. This result may seem surprising, especially since studies have shown higher job demand for managers [38]. Nevertheless, in our study, the three outcomes measured relate to the employee’s perception of his or her own activity. In particular, a manager is generally expected to take on management tasks in addition to their own, unlike an employee. However, since these additional tasks are an integral part of a manager’s job, they may factor into the perceived workload scale. This could explain the absence of a significant difference between employees and managers in their perceived job demand. However, managers reported having slightly higher job control and job support than employees. The strong empowerment (described in the literature as a form of control or autonomy) of managers has been shown previously in nurses [39,40], with a positive correlation between empowerment and positive work-related outcomes [39]. The difference in social support between managers and employees is coherent, as managers may receive support from their subordinates in addition to workers of the same or higher grade, improving the perception of their work [41]. Institutions should work to improve decision latitude and social support in the workplace. This should result in a consequent reduction in job strain and isostrain risks for workers, resulting in an improvement in their occupational well-being, especially for employees who are at greater risk.

### 4.2. The Other Factors of Job Strain and Isostrain

Sociodemographic factors may affect perceptions of work. In our study, we found a positive influence of age on job demand and a negative influence on job control and job support. This resulted in a higher risk of job strain and isostrain for people older than 45 years old. Previous studies have shown that, on the contrary, the risk of job strain seems to decrease with age [38] or has no effect [6]. Similarly, we found that seniority was positively associated with job demand and negatively associated with job support, resulting in a higher risk of isostrain for workers with more than 10 years of seniority. This suggests that, over the years, workers feel that the work they are asked to do becomes increasingly demanding, and that they lose the support from their colleagues or their management, thereby increasing their risk of strain.

According to our results, it seems that women perceived higher job demand than men while receiving less job support, which has previously been shown among managers [22]. Thus, women are slightly more at risk of job strain and isostrain than men, as already established in previous studies [24,38]. While the harmful effects of stress at work are well known among women [42], our study also shows that this is a population at risk.

Our results confirm that personal factors can alter the perception of work. This reinforces the idea that the Karasek model should not be applied without considering the sociodemographic attributes of workers. Institutions should strive to ensure that older (and more senior) workers feel as regarded as younger workers. They must also be aware of potential differences in the perception of work between men and women, in order to take all the measures to enable their employees, regardless of their sex, to feel on an equal footing.

### 4.3. Limitations

We acknowledge some limitations in our study. Despite the fact several factors such as education or shift work are known to have an impact on job strain [24], they are not retrieved by Wittyfit. The use of self-reported responses to questionnaires could have led to over- or under-estimated values, which might lead to a measurement bias. Additionally, the use of a single VAS to assess each dimension of Karasek’s model, instead of a multi-item questionnaire, may reinforce the idea of an affective bias. However, the quadrant classification of the JDCS model, based on historically studied thresholds, allowed us to obtain figures in agreement with the literature, with these values being very close to our first quartile (Q1) and tercile (T1) values (workload: Q1 = 22.5, T1 = 23.9; autonomy: Q1 = 65.8, T1 = 73.7; ambiance: Q1 = 20.0, T1 = 21.0), often used as cut-offs to detect job strain and isostrain risk in various studies [43]. With this construct, we found that 24% of the population was in a job strain situation and 13.1% in an isostrain situation, in line with the literature [9,11,24]. Further, as an ever-expanding, real-world database, we can expect further investigations and comparisons of these results, in the future, with a larger dataset, even with repeated measurements and thus with more measurements per individual. Numerous companies were included in the study, but they only represented a small sample of possible business types. Nevertheless, the use of mixed models allowed us to take into account a “company” effect and to separately estimate the effects of sociodemographic characteristics. As Wittyfit’s clients did not provide further details about their employees, such as their job, our sample might be non-representative of French workers. However, the use of the “Wittyfit” database allowed us to work with a large sample representative of various management positions [30], which was our main focus. These considerations allow us to assume the generalizability of our results.

## 5. Conclusions

Managers have a lower risk of job strain and isostrain. Indeed, they have higher levels of control and support at work than employees. Older workers and those with higher seniority may also be slightly more at risk of job strain. Irrespective of job position, acting on the main dimensions of Karasek’s model (job demand, job control, and job support) at work can be beneficial in improving the well-being of workers. Thus, implementing strategies to improve, among other things, decision latitude and social support in the workplace could help companies reduce the risk of job strain and isostrain.

## Figures and Tables

**Figure 1 ijerph-19-02153-f001:**
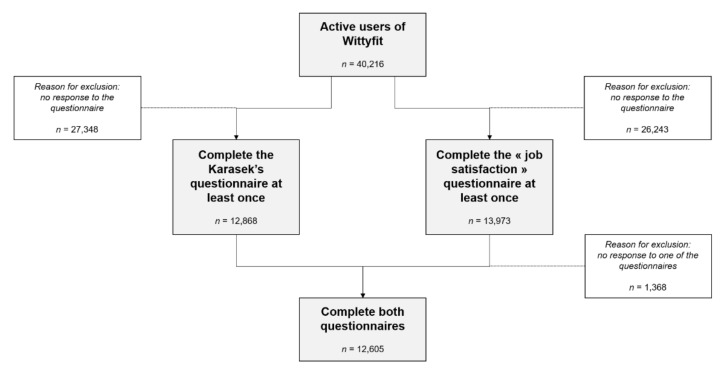
Flow chart of Wittyfit’s users.

**Figure 2 ijerph-19-02153-f002:**
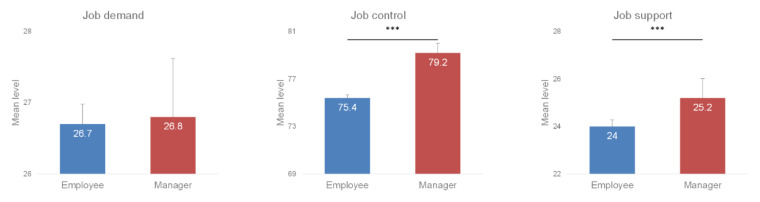
Job demand, job control, and job support by position. Legend: *** *p* < 0.001.

**Figure 3 ijerph-19-02153-f003:**
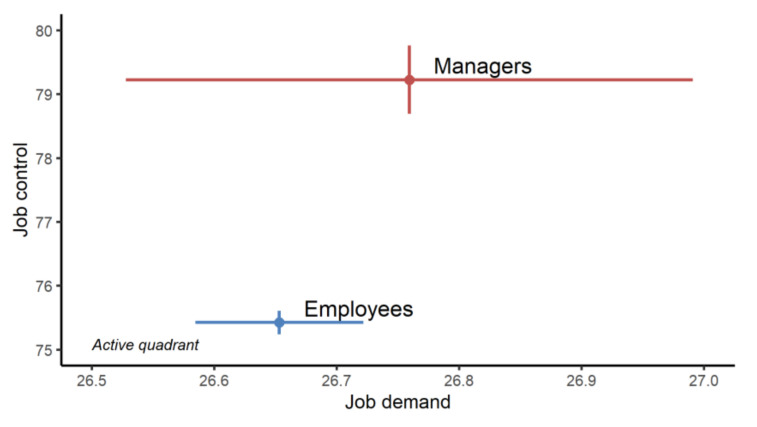
Karasek’s Job Demand-Control-Support model. Point ranges are expressed as means and standard errors. Job demand varies from 9 to 36, job control from 24 to 96.

**Figure 4 ijerph-19-02153-f004:**
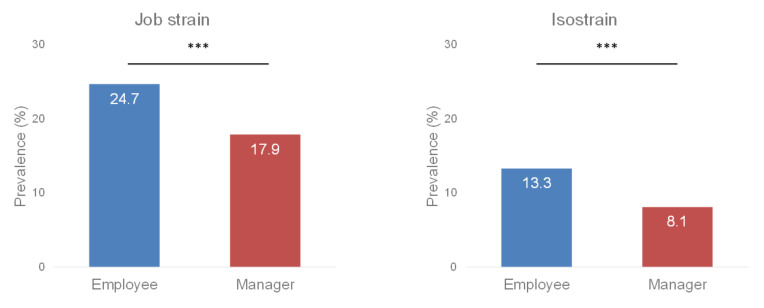
Prevalence of job strain and isostrain by job position. Legend: *** *p* < 0.001.

**Figure 5 ijerph-19-02153-f005:**
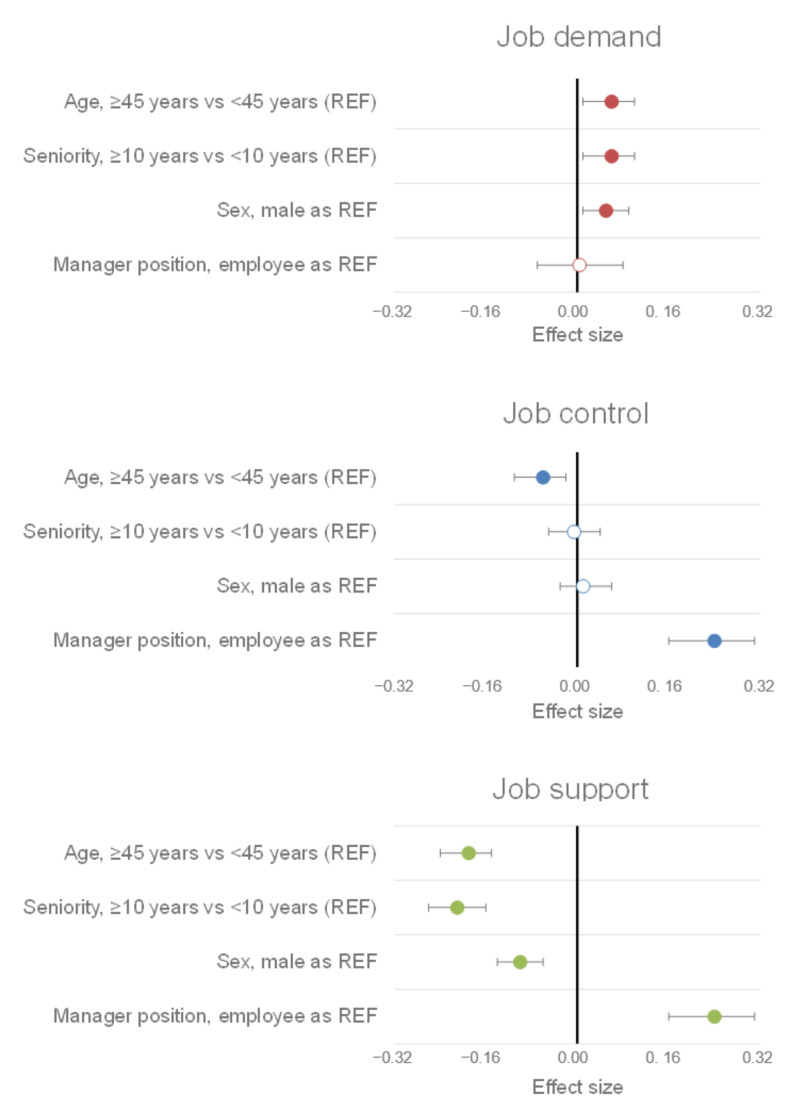
Sociodemographic factors of the Karasek model dimensions. A colored circle means that the measured effect is significant. The reference classes (REF) of the different parameters are the following: age: workers under 45; seniority: workers with less than 10 years of service; sex: male; job position: employee.

**Figure 6 ijerph-19-02153-f006:**
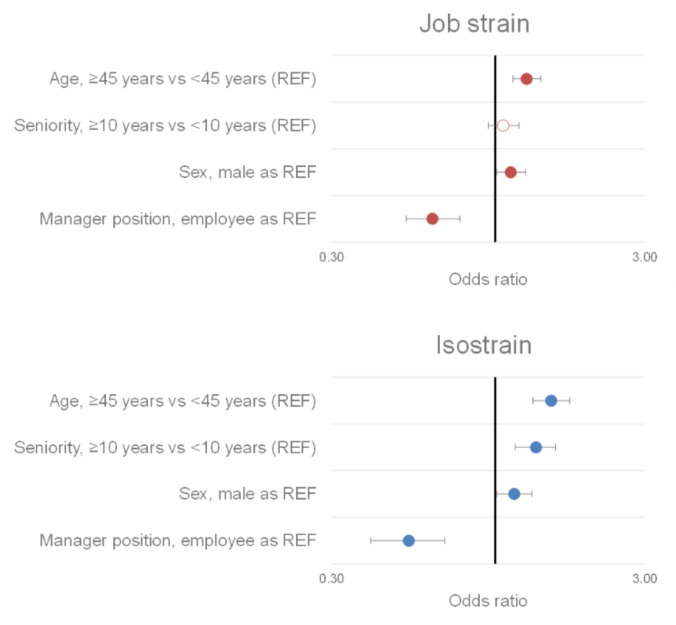
Sociodemographic factors of job strain and isostrain. A colored circle means that the measured effect is significant. The reference classes (REF) of the different parameters are the following: age: workers under 45; seniority: workers with under less than 10 years of service; sex: male; job position: employee.

**Table 1 ijerph-19-02153-t001:** Characteristics of workers according to their classification using Karasek’s Job Demand-Control-Support model (JDCS).

Variable	JDCS Model Groups				*p*-Value
	Active	High Strain	Low Strain	Passive	
JDCS (mean ± SD)					
*n* (%)	5537 (59.8%)	2235 (24.1%)	927 (10.0%)	558 (6.0%)	
Job demand	29.3 ± 4.2	27.3 ± 4.5	15.9 ± 3.4	15.8 ± 3.6	<0.001
Job control	85.1 ± 7.7	55.3 ± 11.2	84.0 ± 8.0	50.5 ± 13.6	<0.001
Job support	25.2 ± 5.6	21.8 ± 6.1	25.2 ± 5.9	21.0 ± 6.7	<0.001
Age (years), *n* (%)					
<45	2829 (51.1%)	1024 (45.8%)	523 (56.4%)	293 (52.5%)	<0.001
≥45	2708 (48.9%)	1211 (54.2%)	404 (43.6%)	265 (47.5%)	
Seniority (years), *n* (%)					
<10	3229 (58.3%)	1238 (55.4%)	638 (68.8%)	369 (66.1%)	<0.001
≥10	2308 (41.7%)	997 (44.6%)	289 (31.2%)	189 (33.9%)	
Sex, *n* (%)					
Male	3349 (60.4%)	1302 (58.3%)	644 (69.5%)	364 (65.2%)	<0.001
Female	2191 (39.6%)	933 (41.7%)	283 (30.5%)	194 (34.8%)	
Position, *n* (%)					
Employee	5041 (91.0%)	2097 (93.8%)	826 (89.1%)	524 (93.8%)	<0.001
Manager	91 (9.0%)	93.8 (6.2%)	101 (10.9%)	34 (6.1%)	

## Data Availability

Data from Wittyfit cannot be transmitted without the prior consent of the company’s corporate clients, except to the University Hospital of Clermont-Ferrand, France, which may use the data for research purposes.

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
