# Peer review of "The Protective Role of Job Control/Autonomy on Mental Strain of Managers: A Cross-Sectional Study among Wittyfit’s Users"

_ijerph, 2022, doi:10.3390/ijerph19042153_

Round 1
Reviewer 1 Report
it was really a pleasure to be able to read and participate in the review of this article. In my personal opinion, the study is well done, methodologically, and results seem really important for the JD-R model. The paper is almost finished. Anyway, I really think that it can profit from some specific suggestions.
1) The title. In my opinion, the current title does not focus on the measured variables or does not explain anything new from the results. Please, try to create a more significant title (even from an originality point of view).
2) The paper is based on the Job-Demand-Control-Support model of Karasek, presented as the gold standard to assess perceptions at work, but two aspects are totally missing: A) the JD-R model is not presented correctly and comprehensively to the reader, it is not presented in such a way as to understand the relationship between variables, it is not used to talk about the psychological mechanisms underlying the model. B) There is a complete lack of a theoretical link between JD-R and the literature on managers, and there is no mention in any way of the mechanisms underlying psychological processes, or of the reasons why the JD-R model should "change" in managers. Why should managers have different values of JD or JR, based on what literature or based on what other results exactly?
3) It is not clear if the two samples are independent (the starting differences of the samples (manag. vs workers) for the SD variables are not presented. Why? These differences could be responsible for the differences in the model. indeed, given the imbalance between the sample of workers and that of managers, it appears that a single data collection was made and then the sample was divided between workers and managers. This ambiguity is even more evident because the authors have not made it clear what defines a manager and what a worker. How were the two champions really divided? Based on what variable? Role only? Was the variable declared by the participants or was it measured by the researchers on the basis of the job profile?
4) You wrote that "Several SD factors like gender, lower education, shift work, are known to have an impact on job strain among workers". So why did you not have a deep analysis of these aspects? there are too few measured variables and therefore each contribution can enrich the description of the results.
5) the article does not describe the theoretical implications of the results on the Karasek or JD-R model. This is a big issue.
6) The article does not explain any practical implication of the results.
congratulations for the article, it can really be very important in the literature if well revised in the literature and especially in the theoretical contribution.
Reviewer 2 Report
This study examined the perception of work among 769 managers, and quantify their risk of job strain and isostrain mainly by comparing with 8488 employees. Although I applaud the authors’ attempt to use the Job-Demand-Control-Support model of Karasek to meet their aim by a large sample, there are some concerns which suggest areas for further improvement and expansion.
- My major concern about this study is that the theoretic contribution is limited. At present, I’m unclear how these findings can contribute to the Job-Demand-Control-Support model of Karasek or relevant literature. Moreover, the results seem to have few additional practical implications. Therefore, I suggest the authors could look at their findings or data in depth and reconsider their contribution of this study.
- My second major concern is about the measurement. First, the authors should present their items used for better understanding. Actually, only three items were used, one item for each variable, in this study. Second, I wonder if it is reasonable to evaluate job demand and job control with the two VAS from the “stress” questionnaire and assess job support by the “ambiance” VAS from the “job satisfaction” questionnaire. These constructs seem to be different, and thus the authors need to make justifications for their measurement. Finally, although the authors acknowledge their use of a single VAS to assess each dimension of Karasek's model rather than a multi-item questionnaire, the measurement bias may have already affected the reliability of conclusions. In addition, on p.2, line 91-92, the authors need to explain the meaning of “values of 21, 70 and 23” more specifically.
- The introduction section seems to be too simple. It would be better if the authors detail more in their review of past literature. For example, they may give definitions and illustrations of job demand, job control and job support.
- In discussing the other factors of job strain and isostrain, the authors seem to only summarize their findings. Therefore, I suggest the authors give explanations and discuss the findings in depth, especially for the inconsistent ones. Moreover, the authors could emphasize the theoretic contributions and practical implications of their study in the discussion section more specifically.
- There are some minor mistakes in the manuscript. For example, on p. 4, line 139-140, the authors stated a non- significant difference on job demand, but they marked significance “***” in figure 2. On p.8, line 217-218, “Finally, we found that women were very slightly more at risk of job strain and isostrain than men”. The word “very” could be deleted in the sentence.

Round 2
Reviewer 1 Report
The authors provided the right responses and integrations for the paper. Congratulations.
Reviewer 2 Report
I’m glad to see that the manuscript has been much improved. However, I still have some concerns which I will outline as follows:
My first concern is still about the measurement. The authors have not yet given sample items of the three outcome variables in the revised manuscript. Therefore, it is still unclear whether these items are representative of the investigated construct, especially for job support. Although the authors argued that job support was assessed by the “ambiance” VAS from the “job satisfaction” questionnaire, designed to assess social support at work (Dutheil et al., 2017), I noticed that the cited reference, “WittyFit-Live Your Work Differently: Study Protocol for a Workplace-Delivered Health Promotion”, comes from the authors’ own team. Thus, it could not sufficiently justify the measure of job support with the “job satisfaction” questionnaire. Furthermore, the authors claimed they added this possible measurement bias to limitations and argued that it comes from self-reported responses. Nevertheless, the measure bias I pointed out is about whether these measures can represent the construct of the variables. Therefore, I suggest more work need to be done to justify their measurements.
Second, on page 5, the authors found that there is no significant difference in job demand between managers and employees (26.8 ± 6.4 vs 26.7 ± 6.3, p = 0.3). However, on page 7, lines 190-192, the authors stated that “The main results show that despite a higher work demand, managers were less subject to job strain and isostrain risks than employees, due to higher job control and support”. Actually, work demand of managers is not significantly higher than that of employees. Thus, “despite a higher work demand” could be deleted.
